# Dynamic Expression of Interferon Lambda Regulated Genes in Primary Fibroblasts and Immune Organs of the Chicken

**DOI:** 10.3390/genes10020145

**Published:** 2019-02-14

**Authors:** Mehboob Arslan, Xin Yang, Diwakar Santhakumar, Xingjian Liu, Xiaoyuan Hu, Muhammad Munir, Yinü Li, Zhifang Zhang

**Affiliations:** 1Biotechnology Research Institute, Chinese Academy of Agricultural Sciences, Beijing 100081, China; liuxingjian@caas.cn (X.L.); huxiaoyuan01@caas.cn (X.H.); 2Division of Biomedical and Life sciences, Faculty of Health and Medicine, Lancaster University, Lancaster LA1 4YG, UK; d.santhakumar@lancaster.ac.uk

**Keywords:** Transcriptome, interferon lambda, ISGs, RNA-Seq, antiviral pathway

## Abstract

Interferons (IFNs) are pleiotropic cytokines that establish a first line of defense against viral infections in vertebrates. Several types of IFN have been identified; however, limited information is available in poultry, especially using live animal experimental models. IFN-lambda (IFN-λ) has recently been shown to exert a significant antiviral impact against viral pathogens in mammals. In order to investigate the in vivo potential of chicken IFN-λ (chIFN-λ) as a regulator of innate immunity, and potential antiviral therapeutics, we profiled the transcriptome of chIFN-λ-stimulated chicken immune organs (in vivo) and compared it with primary chicken embryo fibroblasts (in vitro). Employing the baculovirus expression vector system (BEVS), recombinant chIFN-λ3 (rchIFN-λ3) was produced and its biological activities were demonstrated. The rchIFNλ3 induced a great array of IFN-regulated genes in primary chicken fibroblast cells. The transcriptional profiling using RNA-seq and subsequent bioinformatics analysis (gene ontology, differential expressed genes, and KEGGs analysis) of the bursa of Fabricious and the thymus demonstrated an upregulation of crucial immune genes (viperin, IKKB, CCL5, IL1β, and AP1) as well as the antiviral signaling pathways. Interestingly, this experimental approach revealed contrasting evidence of the antiviral potential of chIFN-λ in both in vivo and in vitro models. Taken together, our data signifies the potential of chIFN-λ as a potent antiviral cytokine and highlights its future possible use as an antiviral therapeutic in poultry.

## 1. Introduction

Viral pathogens pose significant threats to the poultry industry around the globe. This necessitates the development of novel and alternative antiviral therapies to contain the impacts of pathogens. Avian influenza viruses (AIVs) are a particular threat, which cause severe damage to the poultry industry, especially in developing countries where huge monetary losses are incurred [1,2]. Public health is also threatened by AIVs, owing to their zoonotic importance. Active preventive strategies would minimize the risk of viral transmission to humans and also benefit the poultry industry. 

Interferons (IFNs) are pleiotropic functional cytokines with antiviral, antitumor, and natural immune-boosting effects. IFNs play a significant role in eliciting an antiviral state in vertebrates [3]. IFNs are broadly categorized into three distinct types based on their molecular structure, receptor specificity, and induction pathway [4]. Type I IFNs include IFN-α, IFN-β, IFN-ε, IFN-κ, and IFNω, and all signal via common cell surface receptors (IFNαR-1) and (IFNαR-2), which are situated on a broad range of cells [3]. Type II IFNs consist of IFN-γ, which is activated through highly specific ligand interactions with distinct IFN-γ receptors (IFN-γR1) and (IFN-γR2). The third family of IFNs consists of IFN lambda, which interacts with a heterodimeric receptor complex (IL-28Rα and IL-10β). IFN-λ was first discovered in mammals and subdivided into IFN-λ1 (also known as IL-29), IFN- λ2 (IL-28A), IFN- λ3 (IL-28B), and IFN-λ4 [5]. IFNs are crucial in an innate immune response, as their expression and antiviral potential is dependent on their cognate receptor interaction in a particular system [6]. In chickens, type I IFNs primarily interact in fibroblasts, whereas epithelial cells (gastrointestinal and respiratory tract) are the primary site for the actions of type III IFNs [7]. Despite morphological diversity, IFNs share integrated, interconnected, and a precisely coordinated cascade in immunity pathways [3]. 

Ligand recognition and interaction by IFN receptors results in rapid activation of Janus kinase/signal transducers and activators of transcription (JAK-STAT pathway). This leads to phosphorylation of STAT1 and STAT2, activation of interferon stimulated gene factor 3 (ISGF3), binding of IFN-stimulated response elements (ISREs), and expression of IFN stimulated genes (ISGs) [8]. Once expressed, these ISGs demonstrate an essential role in the antiviral response. It is evident from published data that IFNs upregulate identical sets of ISGs, which in turn express antiviral proteins. IFN-induced transmembrane protein (IFITMs), viperin and myxovirus resistance protein (Mx) are some of the potent antiviral proteins expressed in response to viral infections [9]. Once expressed, these ISGs control viral replication, which provides an antiviral atmosphere to limit viral propagation in infected cells.

Compared to the mammalian IFN-λ repertoire (IFN-λ1, IFN-λ2, IFN-λ3, and IFN-λ4), chicken IFN-λ is the sole member in birds and demonstrates structural identity with human IFN-λ3. IFN-λ is chiefly involved in protection against viral infection of the respiratory and gastrointestinal tract epithelia (AIV, NDV, IBV), and due to the distribution of IL-28Rα in epithelium-rich organs, IFN-λ demonstrates significant potential to limit viral propagation [10]. While most of the current studies in chickens are mainly focused on type I and type II IFNs, we investigated the potential of type III IFNs in innate and adaptive immunity.

Previously, it was established that chIFN-α presented a significant delay in the propagation of Rous sarcoma virus and confirmed in vivo [11]. It was also revealed that chIFN-α treatment ameliorates infection progression in experimental chickens with highly pathogenic influenza A virus (HPAIV) subtype H5N1 [12]. Compared to type I IFNs, chIFN-λ has also been shown to elicit moderate antiviral response in both the chicken macrophage cell line HD11 and primary chicken embryo fibroblasts (CEF) [13]. Another published study demonstrated that CEFs treated with recombinant chIFN-λ induced ISGs in a temporal fashion [14]. However, the antiviral potential of chIFN-λ in live animals (e.g., chickens) has not yet been investigated, which could provide evidence for the potential of chIFN-λ in animals per se.

To investigate the impact of exogenous chIFN-λ on the innate immune system in chickens, we first expressed chIFN-λ in a silkworm bioreactor platform utilizing a baculovirus expression vector system (BEVS) [15]. Compared to the *Autographa californica* nucleopolyhedrovirus (AcMNPV)-Sf9 cell expression system, the *Bombyx mori* nucleopolyhedrovirus (*Bm*NPV)-silkworm system possesses greater post-translational modifications and enhanced expression efficiency [16,17]. Comparative transcriptomic profiling revealed the key mechanisms, signaling pathways, and expression patterns of genes involved in interferon-induced immunity. Our results highlight the dynamics of chIFN-λ roles in chicken innate immunity.

## 2. Material and Methods

### 2.1. Cells

Bm5 cells (*Bombyx mori*-derived cell line) were cultured and maintained at 27 °C with 10% fetal bovine serum (FBS, Gibco, USA) in TC100 (insect cell culture medium) (Applichem, Darmstadt, Germany) as per the published literature [18]. For co-transfection, Bm5 cells were cultured at a constant density of 1 × 10^6^ cells per well in six well plates for 12 hours with TC100 media containing FBS. TC100 media without FBS was used to wash the cells twice and a mixture of transfection and co-transfection was introduced to cells. Between 4–6 h post-transfection, FBS was introduced to the cell culture media. For viral amplification and expression, cells were infected with a multiplicity of infection (MOI) of 0.1 for 1–2 h.

### 2.2. Data mining and Bioinformatic Analysis of Chicken IFN-lambda (chIFN-λ)

The Ensembl chicken genome database (ftp://ftp.ensembl.org/pub/release-93/fasta/gallus_gallus/dna/) was extensively screened for homologues of chIFN-λ by employing the BLAST algorithm (http://www.ncbi.nlm.nih.gov/BLAST/). A stretch of sequences demonstrating high sequence identity was identified and characterized. All sequences (avian and mammalian IFN-λ) including Sus scrofa (pig) [NP_001159962.1], Bos taurus (cattle) [NP_001268830.1], Homo sapiens (human) [AAN86127.1], Mus musculus (mouse) [NP_796370.1], Gallus gallus (chicken) [XP_015144667.1], and Xenopus tropicalis (frog) [NP_001165236.1] were acquired from the National Center for Biotechnology Information (NCBI) and aligned using the ClustalW program, and phylogenetic analysis was performed using the neighbor-joining method with bootstrap n = 1000 in MEGA software (version 7). Amino acid sequences of IFN-λ from multiple species were aligned using the ClustalW algorithm. The ESPript 3.0 (http://espript.ibcp.fr/ESPript/cgi-bin/ESPript.cgi) was utilized to analyze the sequences.

### 2.3. Expression of Recombinant chIFN-λ3 (rchIFN-λ3)

In our previous study, we developed a novel defective-rescue recombinant *Bombyx mori* Bacmid (reBmBac) expression system [15]. We used this in-house built and developed system to express chIFN-λ. The reBmBac-silkworm expression system was employed to construct chIFN-λ (interferon lambda-3 [*Gallus gallus*]; Sequence ID: XP_015144667.1; Length: 186). Briefly, in order to enhance expression efficiency by codon optimization, chIFN-λ genes were optimized for expression in the silkworm (*Bombyx mori*) and synthesized by GenScript Company (China). Plasmid-containing ORF1629+ with gene of interest (chIFN-λ) and Pph as a promoter was co-transfected with reBmBac in the Bm5 cell line. Recombinant virus containing the chIFN-λ gene was harvested 4–5 days post co-transfection. Expression product was acquired after 4–5 days of silkworm/pupae infection. The plaque assay was performed to evaluate the recombination efficiency [18]. Luciferase assay kit (Promega, USA) was employed to analyze expression quantity of luciferase in 50 μg of protein lysate. The Bradford method was used to measure the amount of protein [19]. Antiviral activity of chIFN-λ was assayed in the GFP-reduction assay using recombinant vesicular stomatitis virus (VSV-GFP) [20].

### 2.4. Preparation of Primary Chicken Embryo Fibroblast

CEFs were prepared from 9–11 days old specific pathogen free (SPF) chicken eggs and maintained in cell culture flasks [21]. After 24 hours, CEFs were stimulated with chIFN-λ and cells were harvested after 12 hours post treatment, snap frozen, and stored at −80 °C for further processing. All experiments were performed in triplicate.

### 2.5. Birds and Management

The present study was conducted in accordance with animal ethics guidelines and approval was given by the Beijing Administration Office of Laboratory Animals, China. A total of 60 newly hatched SPF chicks were obtained from Beijing Arbor Acre Company Ltd., P.R. China. Chicks were reared in cages (n = 10 birds/cage) and placed in six cages in a temperature-controlled environment at the Biotechnology Research Institute, Chinese Academy of Agricultural Sciences (CAAS), P.R. China. Birds were offered standard commercial feed obtained from CP Group Ltd., P.R. China. Unrestricted access to water was provided via nipple drinker lines and *ad libitum* feed was offered. A treatment group of 14-day old chicks were injected daily with chIFN-λ (10,000 IU/kg body weight) (105 IU/mL). Phosphate buffer saline (PBS) was injected intramuscularly to the control group. The bursa of Fabricious and thymus were obtained by euthanizing the chickens at five days post-treatment. Tissue samples were rapidly collected, snap-frozen in liquid nitrogen, and stored at −80 °C for further processing.

### 2.6. RNA Extraction and Sequencing

Total RNA was extracted from virus-infected or mock-treated CEFs (in triplicates), as per manufacturer’s guidelines [22]. Similarly, a total of five immune organs (bursa of Fabricious and thymus) were pooled (in duplicates) from randomly selected chicken from each virus- or mock-infected group. Total RNA extraction was performed as per manufacturer’s instructions [22]. Extracted RNA quality was analyzed by employing 1% agarose gel and RNA integrity was assured using RNA Nano 6000 Assay Kit from Bioanalyzer 2100 System (Agilent Technologies, CA, USA). Extracted samples were sent to Novogene Beijing for sequencing. Samples were sequenced using HiSeq X Ten (Ilumina) and PE150 platforms.

### 2.7. RNA-Seq Quality

RNA-seq generated from CEF, bursa of Fabricious and thymus samples of chicken (both chIFNλ-treated and control groups) are presented in Appendix A. Reads were mapped to the reference genome database (ftp://ftp.ensembl.org/pub/release-89/fasta/gallus_gallus/dna/). Individually mapped reads for each sample were assembled by StringTie (v1.3.3b) using a reference-based approach. FeatureCountsv1.5.0-p3 was utilized to estimate read numbers mapped to each gene. Fragments per kilo base of transcript sequence per million base pairs sequenced (FPKM) of each gene was analyzed on the basis of length of gene and read count mapped to this gene. Differential expression analysis was accomplished by employing DESeq2 R package (1.16.1). Using Benjamini and Hochberg’s approach, *p*-values were adjusted for controlling false discovery rate (FDR). Genes with (*P* < 0.05, │log2fold change│>1) observed by DESeq2 were designated as differentially expressed.

### 2.8. Gene Ontology (GO) and KEGG Enrichment Analysis

For differentially expressed genes, both gene ontology (GO) enrichment analysis and Kyoto Encyclopedia of Genes and Genomes (KEGG) pathway enrichment was conducted using the ClusterProfiler R package. GO terms with adjusted *p*-values < 0.05 were considered as significantly enriched (http://www.genome.jp/kegg/).

## 3. Results

### 3.1. Bioinformatic Analysis of chIFN-λ

Using the chicken IFN gene as a query, we constructed the phylogenetic tree by employing the neighbor joining method (bootstrap n = 1000). This demonstrates the relationship of chIFNλ with its mammalian orthologues by illustrating that chIFN-λ is distinct in its evolution. Furthermore, this revealed the contrasting consensus sequence from databases including Ensembl and Genbank. chIFN-λ encodes a putative protein of 186 amino acids and further demonstrates typical characteristics of type III IFNs. A pairwise BLAST analysis demonstrated that chIFN-λ shares 36%, 34%, 39%, 34% and 33% sequence similarity with recently characterized pig, mouse, human, cattle, and frog IFN-λ, respectively. Based on amino acid homology, conserved amino acids among distinct avian and mammalian IFN-λ are identified. Taken together, this comparative characterization further shows that chIFN-λ shares characteristic features of type III IFNs (Appendix A).

### 3.2. Expression of chIFN-λ in Baculovirus Expression Vector System (BEVS)

In order to construct chIFN-λ, we employed a BEVS study. In order to determine the expression efficiency, we used a luciferase reporter gene for quality control as we described previously [15]. The luciferase gene was acquired from pGL3-Basic vector by employing BglII/XbaI digestion and insertion into the BamHI/XbaI-digested pVL1393 vector to construct pVL1393-luc vector. A combination of pVL1393-luc and reBmBac DNA was co-transfected in Bm5 cells (Figure 1). A viral plaque assay was used to determine a suitable virus strain with which to express luciferase. Supernatant from Bm5 cells containing recombinant *Bm*NPV (reBm-luc) was harvested five days post-transfection before inoculation into silkworms. After four to five days, protein was harvested from silkworms and 50 μg protein from lysed larval haemolymph was subjected to luciferase assays. Luminescence detected from silkworm larval haemolymph was approximately 3.42 ± 0.52 × 10^8^ relative light units (RLU), compared to 150–300 RLU from luc-negative virus-infected samples. PCR amplification (qPCR) further verified and validated the chIFN-λ gene expression in BEVS (Appendix A).

### 3.3. Characterization of chIFN-λ-induced Gene Expression in Chicken

In order to investigate the possible biological, cellular, and molecular mechanisms involved in the cascade of interferon-induced immunity, we performed transcriptomic analysis on chicken embryo fibroblasts and organs of live chickens. Transcriptomes from the bursa of Fabricious and thymus (most important immune organs in chicken) were compared with the control group to identify differentially expressed genes (DEGs) among all groups. Experimentation started at day 14 post-hatch as this is a phase of rapid growth and development, and we hoped to achieve biologically active transcriptional changes. The differences in DEGs observed in the present study control cellular architecture, immune function, metabolic pathway, and muscular function.

It has previously been established that huIFN-λ signals via IL-10 and IL-28R exhibit typeI-like antiviral potential [23]. Protection from simian foamy virus (SFV) and avian influenza (AI) augments the antiviral functioning and further postulates its diverse antiviral potential against avian pathogens. In this context, we stimulated chickens with silkworm-expressed chIFN-λ and profiled the gene expression in immune organs (thymus and bursa) and compared it with that in primary chicken fibroblasts using RNA-Seq. An overall low ISG expression was noticed in chIFN-λ-stimulated CEF; out of 26,616 genes, 161 were DEGs (84 upregulated and 77 downregulated) (*P* < 0.05, │log2fold change│>1) (Figure 2A). Although CEF do not possess receptors for IFN-λ, slight temporal expression of DEGs in response to chIFN-λ treatment signifies its antiviral potential in primary cells.

Next, we monitored the gene expression in the thymus and bursa. Between the chIFN-λ-treated and non-treated thymus, a total of 23,801 genes were expressed. Among them, 331 genes were DEGs, in which 177 genes were upregulated and 154 genes were downregulated (Figure 2B). In the bursa of Fabricious, 289 out of 23,951 genes were differentially expressed (130 upregulated and 159 downregulated) (Figure 2C). Interestingly, a relatively low number of genes overlapped among these three groups (Figure 2D). In order to confirm the expression of DEGs, we used a conventional approach (qPCR) and show (Appendix A) a scenario corresponding to the RNA-seq data. On the basis of abundance and fold change, DEGs were further characterized (Appendix A).

Cumulatively, a significant upregulation of crucial cytokine and chemokine genes (IL1-β, CCL4, CCL5, and CX3CL1) was observed. These are broadly involved in antiviral response, apoptosis, cellular proliferation and differentiation, cytokine–cytokine receptor interaction and inflammation pathways [24] (Figure 3). Due to the induction of a distinct subset of genes, a lower level of antiviral activity is observed as compared to type-I IFNs. It is speculated that the activation of chIFN-λ is similar to type-I IFNs but they are diverse in functional capability. The chIFN-λ have particular significance in viral infections of epithelial origin, where they are optimally active by eliciting a broad antiviral state. Using conventional approaches, we have confirmed the expression of selected genes as shown in Appendix A.

### 3.4. Functional Analysis of DEGs

DEGs were further analyzed for GO terms and the KEGG pathway by utilizing DESeq2 [25]. Of 956 GO terms associated with chIFN-λ-treated CEF, 112 GO terms were significant (*P* < 0.05) (Figure 4A). In the bursa, among biological processes, we observed the Wnt signaling pathway (WIF1/CAMK2A), cytokine–cytokine receptor interactions (TNFSF11), the apelin signaling pathway (RYR2/MYL4), and the significant antiviral pathway (novel gene) in cellular components (Figure 4B). In the thymus, out of 1712 GO terms, we observed 309 significant, and in the bursa, out of 2298 GO terms, 637 were significant (*P* < 0.05). In order to understand the biological functions associated with DEGs, we further analyzed the data in three distinct categories, including biological processes (BP), cellular components (CC), and molecular functions (MF) (Figure 4C).

### 3.5. KEGG Pathway Enrichment

Further to gene ontology and differential expression, we investigated KEGG pathway enrichment. In CEFs, significant enrichment was seen in pathways including the MAPK signaling pathway (FOS/IL1B/FOSB), the toll-like receptor signaling pathway (FOSB, IL8L1, IL1B, FOS, CCL5), influenza A (IL8L1/IL1B/CCL5), cytokine–cytokine receptor interactions (CCL20/IL8L1/IL1B/CX3CR1/CCL5), salmonella infection (FOSB/IL8L1/IL1B/FOS), the NOD-like receptor signaling pathway (IL8L1/IL1B/CCL5), and herpes simplex infection (FOSB/IL1B/FOS/CCL5) (Figure 5A). In bursa, Wnt signaling (WIF1), the apelin signaling pathway (RYR2), and the calcium signaling pathway (RYR2) were significantly observed (Figure 5B). For the thymus, the NOD-like receptor signaling pathway (PLCB1/MAPK11), the MAPK signaling pathway (SRF/MAPK11), influenza A (RSAD2/MAPK11), and MAPK11 (salmonella, toll-like, herpes simplex infection) were observed (Figure 5C). Collectively, apoptosis (JUN/BIRC5/CTSC/ACTG1), RNA degradation (ENO1/BTG2/C1D), the TCA cycle (MDH1/IDH3A), the p53 signaling pathway (PERP1/CCNB2), biosynthesis of amino acid (ENO1/IDH3A), influenza A (RSAD2/JUN/ACTG1), and the toll-like receptor signaling pathway (JUN) were among the most significant.

## 4. Discussion

Here, we present the first comprehensive report on cloning and expression of chIFN-λ by employing BEVS and demonstrate that it is biologically active in both CEF (in vitro) and live chickens (in vivo). The identification of this potentially significant IFN among the IFN family advances fundamental aspects and functionality of chIFN-λ in avian type-III IFNs. It is evident from the data that this IFN, like human interferon lambda (HuIFN-λ), demonstrates similar type-I IFN-like properties. However, a distinct pattern of expression of ISGs in chIFN-λ contrasts it from other type-I IFNs. Knowledge regarding IFNs is fundamental as rapid outbreaks of viral pathogens cause huge economic losses to the poultry industry every year. The present study investigates the ISGs and signaling pathways associated with avian immunity and will bring new horizons to target problematic viral pathogens, e.g., AIVs, circulating within the poultry industry.

Interferon lambda is a biologically active type-III interferon which primarily acts on epithelial tissues [3]. Studies have demonstrated the antiviral potential of IFN-λ against highly pathogenic avian influenza by eliciting a broad antiviral state [10]. IFN-λ is structurally peculiar as it possesses five exonic regions located on chromosome 7, contrary to type-I IFNs, which are intronless and situated on the Z sex chromosome in chicken [5,26]. This is in agreement with human IFN-λ subfamily which are anatomically identical by possessing five exonic regions on chromosome 1 of the human genome [5]. Furthermore, 36% of amino acids are identical between HuIFN-λII and chIFN-λ, which signifies the similarity of these two IFNs. However, unlike mammals, only one member exists in chicken (chIFN-λ). This is in agreement with the other types of chicken IFNs, which have fewer members compared to mammalian IFNs [27].

Reduced expression of ISGs in response to chIFN-λ in our experiment demonstrates the fact that CEFs are optimally less receptive to IFN-λ, which is in agreement with published reports [10]. One study revealed that chIFN-λ can actively inhibit the viral replication of AI in primary embryonic tracheal organ cultures and CLEC-213 (chicken lung cell line). It is further postulated that with treatment of chIFN-λ, ISGs are expressed significantly, especially Mx gene, which is primarily expressed in epithelial rich organs (i.e., trachea, lungs, and intestine) was also observed in the present study [10]. Furthermore, studies have also revealed that a high degree of cell type specificity in receptor–ligand interactions make avian IFNs distinct from mammalian IFNs. Recently, it has been established that chicken IFN-λ inhibits low pathogenic influenza virus replication in CEFs; however, as compared to chIFN-γ and chIFN-β, higher doses are required to induce ISGs and maintain the strong antiviral state in the cells [14]. GO and KEGG analysis of each experimental group demonstrated overlapping biological functions. An important gene involved in the host response of infected samples is RSAD2, also termed viperin, which is one of the potent interferon stimulated genes (ISGs) responsible for eliciting a broad antiviral state against a variety of viral and bacterial pathogens [28]. In mammals, it is highly expressed in response to invading viral infections [29]. Elevated expression of viperin in chIFN-λ-treated organs further augments the expression of ISGs in response to injected IFN in vivo. Viperin was upregulated in response to chIFN-λ treatment, which is symbolic for all ISGs. IFN-inducible transmembrane protein-1 (IFITM-1) is one of the potent ISGs expressed in response to either type of IFN and plays an antiviral role by blocking cytoplasmic entry [30]. It is further demonstrated that IFITM alters membrane fluidity, hence producing curvature in the outer leaflets of the membrane or by interfering with intracellular cholesterol homeostasis [31,32]. Significant upregulation of IFITM3 in the chIFN-λ-treated thymus augments the temporal expression of ISGs in response to IFN treatment. Further studies are needed to investigate the possible future role of chIFN-λ as a potent and novel therapeutic in the poultry industry.

Although the immune response elicited by type III IFNs is still not very clear, in the present study we also found some novel genes involved in the cascade of the avian immune response. Furthermore, in vitro exposure of CEF to chIFN-λ demonstrated a rapid surge of pro inflammatory cytokines. Considering their vital role in immune pathways, cytokine gene expression is widely employed as an indicator for the immune response. We did observe some genes that were previously illustrated in publications; one such example is chemokine (C-C motif) ligand 1 (CCL1, ENSGALT00000003670) [33]. Chemokines are secreted chemotactic cytokines that play a fundamental role in the recruitment and migration of lymphoid and myeloid cells in target tissues, and hence govern the avian immune response [34]. CCL1 is a chemokine secreted by monocytes that is capable of activating macrophages and T lymphocytes [35]. CCL20, like its mammalian orthologue, is responsible for recruiting lymphoid cells and is involved in the early immune response in chickens [36]. Likewise, CCL1, CCL4, and CCL5 were also upregulated in CEF and are chiefly involved in the innate avian immune response.

The present study describes the transcriptomic analysis of differential gene expression following exposure to chIFN-λ and the resultant pro-inflammatory response in both CEF and chicken tissues. This response ostensibly is due to rapid and sustained signaling via cell surface receptors and a surge of chemokines and cytokines, which in turn create an antiviral environment. A contrasting feature of the present study is the upregulation of the toll-like receptor (TLR) signaling pathway in all three treatment groups, where it is evident that numerous genes are upregulated in TLR mediated cytotoxicity. TLR15, a unique chicken receptor expressed on the surface of fibroblasts, heterophils, and macrophages, shares 30% sequence identity with TLR2 [37]. It is evident from experimentation that TLR15 is a broad spectrum TLR that has the capability to recognize heat stable components of both gram-positive and gram-negative bacteria, CpG oligonucleotides, lipopolysaccharide (LPS), and tripalmitoylated lipopeptide [38]. TLR15, an avian-specific TLR, plays a significant role in avian immune responses against bacterial and viral pathogens. Recently, it has been demonstrated that diacylated lipopeptide from *Mycoplasma synoviae* activated TLR15 and regulated innate immune responses [39]. Similarly, significant upregulation of TLR15, observed in the present study, highlights a possible role of chIFN-λ against *Mycoplasma* infections in chicken. However, it warrants future studies to delineate the molecular processes.

It has also been established by repeated experimentation that chIFN-λ has been seen to cause delay in viral excretion and the spread of highly pathogenic avian influenza (HPAI) H5N1 [10]. It is evident that in mammals, IFN-λ elicits a protective antiviral response toward AI, whereas IFN-λ plays a minor role in lung epithelia [40]. Similarly, in the respiratory tract of chickens, not all mucosal cells are responsive to chIFN-λ. Therefore, treatments can only delay, but do not significantly support the complete removal of viral loads of H5N1 or halt the virus crossing the epithelial barrier [41]. However, for low-pathogenic avian influenza (LPAI), it is evident that chIFN-λ has demonstrated significant antiviral activities [42]. Recent reports revealed another contrasting feature of IFN-λ, where it significantly elicited strong antiviral potential on intestinal epithelial cells to control murine rotaviruses [43,44]. It will be fascinating to investigate in the future whether the same antiviral phenomena occurs, and chIFN-λ might also demonstrate epitheliotropism like rotaviruses and halt viral pathogens of the gastrointestinal tract in chickens.

Nuclear factor kappa-B (NF-KB) is the most significant, evolutionarily conserved, pleiotropic, inducible transcription factor responsible for regulating genetic expression in a variety of fundamental processes, including apoptosis, growth, immune response, inflammation, stress response, etc. [45]. Notably, the upregulation of NF-KB in response to chIFN-λ treatment on CEF signifies their potent role in the immune response. Activator protein 1 (AP-1) is a transcription factor complex highly responsive for cytokine signaling and growth promotion [46]. Formed through noncovalent dimerization between the FOS and JUN family of nuclear oncogenes, this complex activates AP-1-dependent genes, hence controlling cell proliferation, differentiation, and apoptosis [47]. Consistent with these observations, our study demonstrated that many genes associated with this pathway were upregulated. This finding suggests a link between AP1 and the transcriptional cascade associated with recombinant interferon treatment. Overall, transcriptomic analysis revealed significant upregulation of FOS and JUN in CEF and bursa, and thymus of chicken.

The innate immune response is a highly complex, precise, interconnected, and integrated response that relies on many factors. The genes investigated in our study control direct protein interactions and are significantly involved in the avian innate immunity cascade. However, further validation of a broad set of immunity-related genes will also be required to elucidate the mechanism of interferon-induced immunity. A more comprehensive study including a larger set of immune genes and multiple recombinant IFNs, which will correlate their integrated role, will enable researchers to provide comprehensive insight into the avian innate response. Other future studies involving backyard poultry to assess whether similar patterns of innate immunity prevail in indigenous breeds in response to chIFNλ are also important and will further develop our understanding of avian immunity.

## 5. Conclusions

In the current study, we employed RNA-Seq to illustrate vital transcriptomes involved in the cascade of avian biology and observed divergent results in recombinant interferon-treated chickens compared to a control group chickens. Our data suggest that significant antiviral, cell cycle regulators, and biologically active genes are expressed in response to administered chicken IFN. Functional characterization of these vital genes warrants further investigation to determine the future possible role for recombinant chicken IFN in the poultry industry.

## Figures and Tables

**Figure 1 genes-10-00145-f001:**
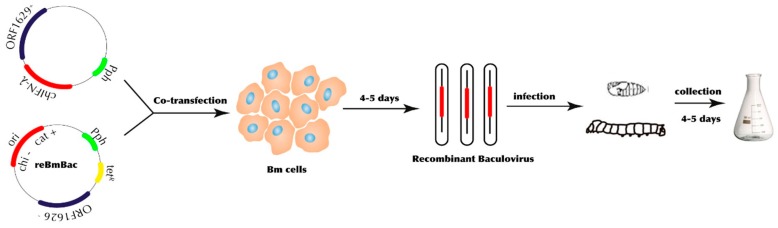
Construction strategy of recombinant baculovirus by employing silkworm expression vector system (BEVS).

**Figure 2 genes-10-00145-f002:**
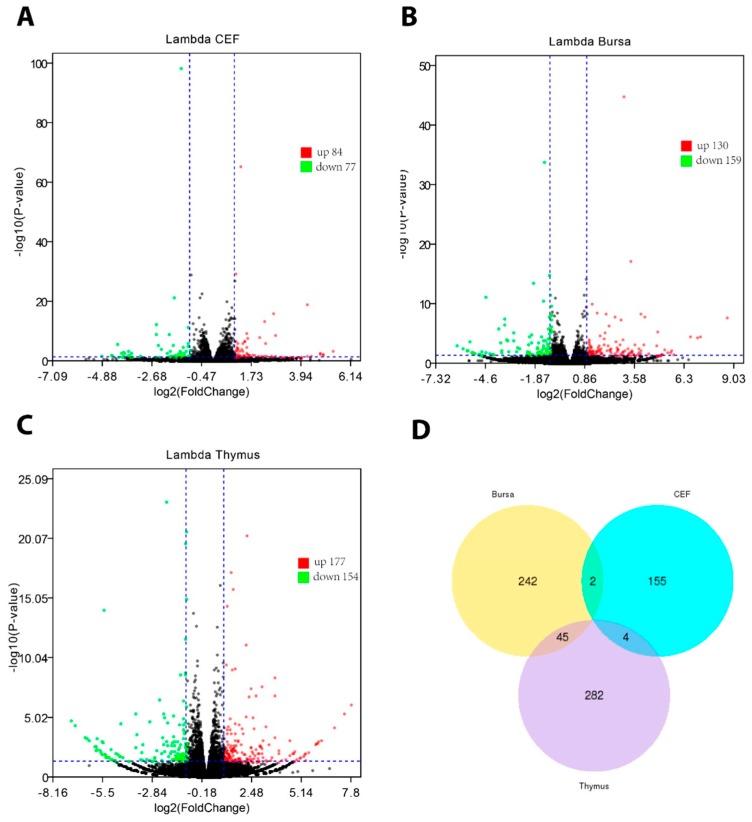
Gene expression representation by volcano plot diagrams and Venn diagram. **A**: Differentially expressed genes (DEGs) in chIFN-λ-treated chicken embryo fibroblasts (CEF). **B**: DEGs in chIFN-λ-treated bursa of Fabricius. **C**: DEGs in chIFN-λ-treated thymus. **D**: Venn diagram representing gene sharing. Red, green, and blue dots represent upregulated, downregulated, and sum of DEGs, respectively. Differential expression patterns demonstrate the temporal expression of genes expressed in the three groups.

**Figure 3 genes-10-00145-f003:**
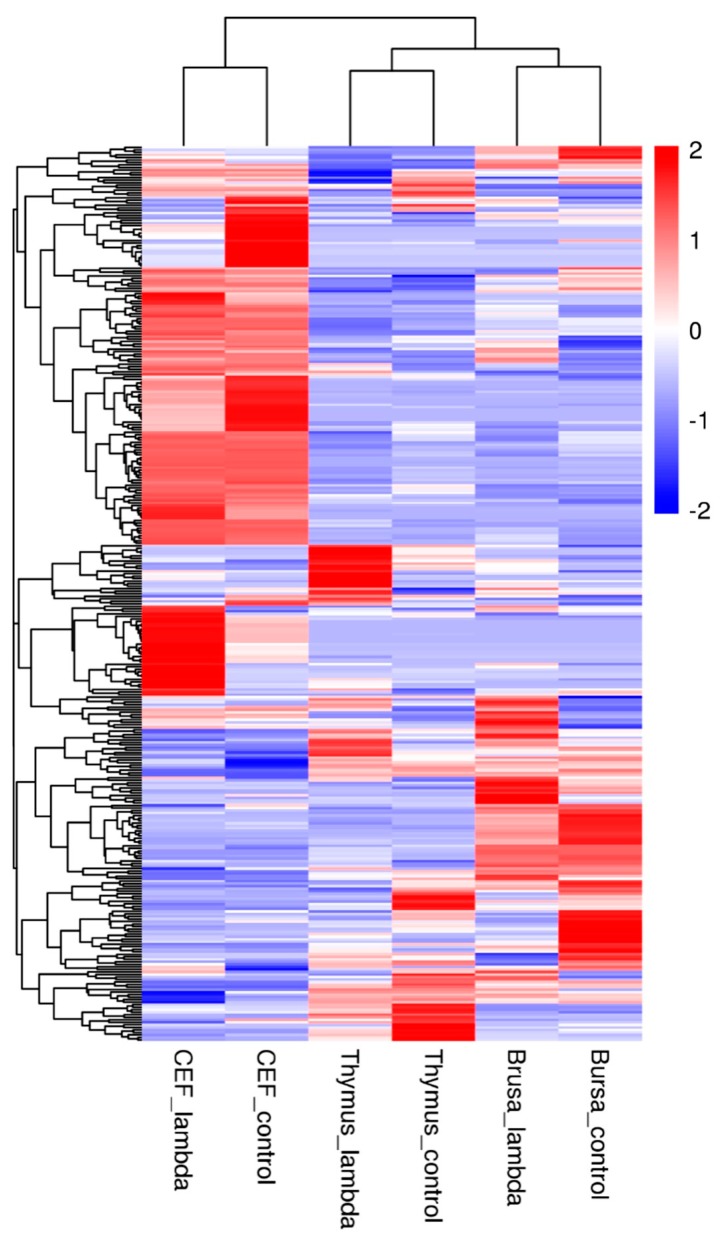
Heat map of DEGs in the CEF, bursa, and thymus. The color bar represents the level of differential expression compared to the control (PBS).

**Figure 4 genes-10-00145-f004:**
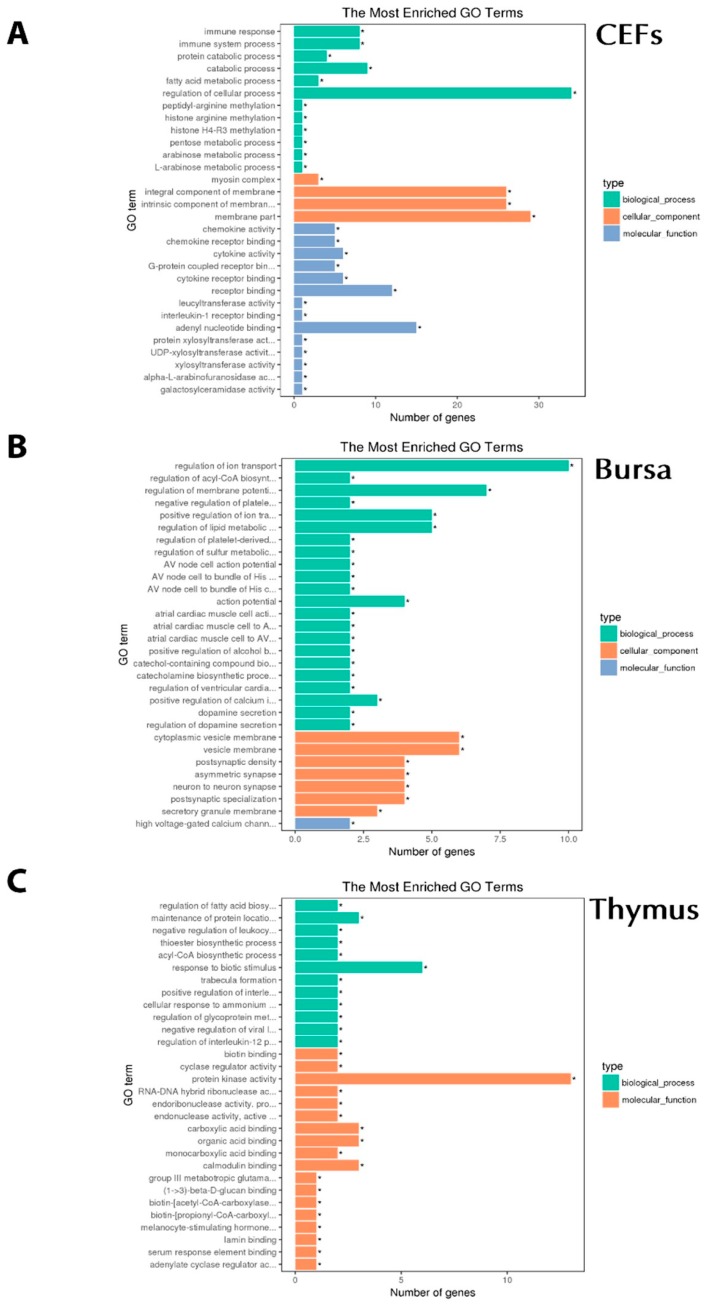
Gene Ontology (GO) analysis associated with chIFN-λ-treated organs. **A**: chIFN-λ-treated CEF. **B**: chIFN-λ-treated bursa of Fabricius. C: chIFN-λ-treated thymus.

**Figure 5 genes-10-00145-f005:**
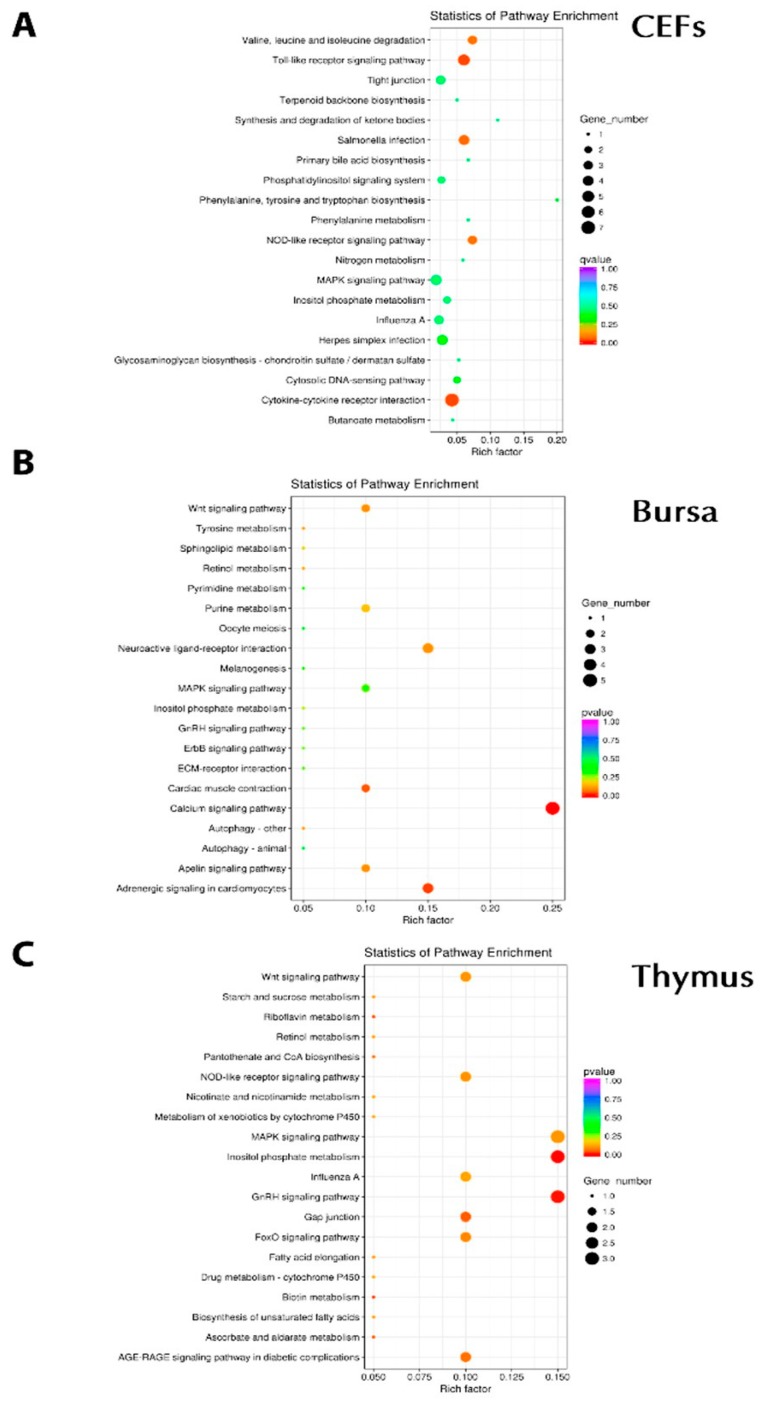
Pathway enrichment associated with chIFN-λ treatment. **A**: chIFN-λ-treated CEF. **B**: chIFNλ-treated bursa of Fabricius. C: chIFN-λ-treated thymus.

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
