# Peer review of "Dynamic Expression of Interferon Lambda Regulated Genes in Primary Fibroblasts and Immune Organs of the Chicken"

_genes, 2019, doi:10.3390/genes10020145_

Round 1

Reviewer 1 Report

Manuscript ID: genes-424618 - Type of manuscript: Article

Title: Dynamic Expression of Interferon Lambda Regulated Genes in Primary Fibroblasts and Immune Organs of Chicken

Authors: Mehboob Arslan *, Xin Yang *, Diwakar Santhakumar, Xingjian Liu,

Xiaoyuan Hu, Muhammad Munir *, Yinü Li *, Zhifang Zhang *

Comments:

This manuscript is describing the expression of IFN-lambda in vivo, using bursa and thymus cells in comparison with CEF in vitro, respectively. The manuscript is well written and there are substantial information for IFN therapy in animals, specifically in poultry.

In the abstract, there are three points who authors should pay attention, that are:

line 16 - please replace "birds" for "poultry" or "chicken" or "avian".

Line 17 - please, as suggestion, the authors should remove "invading" from sentence. In my opinion sounds redundant.

Line 27 - Also, I suggest the authors replace "Intriguingly" for "interestingly".

The introduction is complete and self-explanatory, as well as the material and methods, results and discussion.

Author Response

Response to Reviewer 1 Comments

Point 1: in - line 16 - please replace "birds" for "poultry" or "chicken" or "avian".

Response 1: Thanks for your suggestions. We have now replaced Aves with the "poultry".

Point 2: Line 17 - please, as suggestion, the authors should remove "invading" from sentence. In my opinion sounds redundant.

Response 2: As per kind suggestions of reviewer, we have removed the word ‘invading’ from the sentence.

Point 3: Line 27 - Also, I suggest the authors replace "Intriguingly" for "interestingly".

Response 3: We have replaced it with the word "interestingly".

Reviewer 2 Report

Please see attached file for comments on the text.

Supplementary figures: The title of Supp Fig 2A should be changed to 'Gene expression in IFNlambda induced CEF' and for 2B: 'Gene expression in chicken tissues'

Author Response

Response to Reviewer 2 Comments

Point 1: In line-145 Experimental design needs to be explained more clearly. How many replicates in each group?

Response 1: Thanks for suggestions and we have now elaborated it further and we hope it is clear now.

Point 2: In line-161 Was a fold change cut-off used?

Response 2: We used cut-off (P< 0.05, │log2fold change│>1).

Point 3: In line-196 explain why you studied these 2 tissues in particular?

Response 3: We employed these tissues as these are the immune organs in chicken and play significant role in cascade of avian immunity.

Point 4: In line-298 Re-word this sentence. It doesn’t make sense to me.

Response 4: We have re-written this part according to the respected reviewer’s suggestion.

Point 5: In line-312 which gene?

Response 5: In line-312, the correct name is IFN-inducible transmembrane protein 1 (IFITM1). As per reviewer’s suggestion, it is corrected.

Point 6: In line-342 sentence is incomplete.

Response 6: We are very sorry for writing incomplete sentence. Now it is completed.

Point 7: In line-343 antibacterial role of TLR15?

Response 7: Significant upregulation of TLR15 in response to chIFN-λ might be effective as anti-Mycoplasma infection.

Point 8: In line-389 the supplementary table should order genes by order of fold-change: highest to lowest.

Response 8: Considering the Reviewer’s suggestion, we have generated the table by order of fold change in highest to lowest trend.

Point 9: Supplementary figures: The title of Supp Fig 2A should be changed to 'Gene expression in IFNlambda induced CEF' and for 2B: 'Gene expression in chicken tissues'

Response: Considering the reviewer’s valuable suggestion, we have changed the tiles of the figures as Fig 2A: Gene expression in IFN lambda induced CEF and 2B as Gene expression in chicken tissues.

Point 10: Language improvement of the manuscript.

Response: We are really grateful to the reviewer’s comments, we have accepted all the changes, either insertion or deletion made by the respected reviewer in this manuscript. The manuscript was language edited by the Language Editing Service of MDPI.
